# Maternal prenatal screening programs that predict trisomy 21, trisomy 18, and neural tube defects in offspring

Yiming Chen[1,2]◉*, Wenwen Ning[2]◉, Yezhen Shi[3], Yijie Chen[2], Wen Zhang[1], Liyao Li[1], Xiaoying Wang[1]

1 Department of Prenatal Diagnosis and Screening Center, Hangzhou Women's Hospital (Hangzhou Maternity and Child Health Care Hospital), Hangzhou, Zhejiang, 2 Department of the Fourth Clinical Medical College, Zhejiang Chinese Medical University, Hangzhou, Zhejiang, 3 Data Analysis Department, Zhejiang Biosan Biochemical Technologies Co, Ltd, Hangzhou, Zhejiang, China

◉ These authors contributed equally to this work.
* cxy40344@163.com

**Data Availability Statement:** All relevant data are within the paper and its Supporting Information files.

## Abstract

### Objective

To determine the efficacy of three different maternal screening programs (first-trimester screening [FTS], individual second-trimester screening [ISTS], and first- and second-trimester combined screening [FSTCS]) in predicting offspring with trisomy 21, trisomy 18, and neural tube defects (NTDs).

### Methods

A retrospective cohort involving 108,118 pregnant women who received prenatal screening tests during the first ($9-13^{+6}$ weeks) and second trimester ($15-20^{+6}$ weeks) in Hangzhou, China from January–December 2019, as follows: FTS, 72,096; ISTS, 36,022; and FSTCS, 67,631 gravidas.

### Result

The high and intermediate risk positivity rates for trisomy 21 screening with FSTCS (2.40% and 5.57%) were lower than ISTS (9.02% and 16.14%) and FTS (2.71% and 7.19%); there were statistically significant differences in the positivity rates among the screening programs (all $P < 0.05$). Detection of trisomy 21 was as follows: ISTS, 68.75%; FSTCS, 63.64%; and FTS, 48.57%. Detection of trisomy 18 was as follows; FTS and FSTCS, 66.67%; and ISTS, 60.00%. There were no statistical differences in the detection rates for trisomy 21 and 18 among the 3 screening programs (all $P > 0.05$). The positive predictive values (PPVs) for trisomy 21 and 18 were highest with FTS, while the false positive rate (FPR) was lowest with FSTCS.

### Conclusion

FSTCS was superior to FTS and ISTS screening and substantially reduced the number of high risk pregnancies for trisomy 21 and 18; however, FSTCS was not significantly different

**Funding:** This study was supported by The Joint Fund Project of Zhejiang Provincial Natural Science Foundation of China under Grant (LBY23H200009) and Yiming Chen was chief investigator.

**Competing interests:** The authors have declared that no competing interests exist.

in detecting fetal trisomy 21 and 18 and other confirmed cases with chromosomal abnormalities.

## 1. Introduction

Fetal chromosome aneuploidy is an important issue in human reproductive medicine and an important cause of spontaneous abortion and neonatal congenital malformation [1]. The most common abnormal autosomal aneuploidies are trisomy 21 and 18 [2]. Trisomy 21, also known as Down's syndrome (DS), is a common chromosomal abnormality caused by an increase in the number of chromosome 21 [3]; the incidence of live births with DS is 1/600–800 [4]. The incidence of DS has shown an increasing trend due to the increasing age among pregnant women and the wide application of assisted reproductive technology. Trisomy 18, also known as Edward's syndrome (ES), is another chromosomal disorder caused by the addition of one chromosome 18; the overall incidence of trisomy 18 in live births ranges from 1/2500–2600 [5, 6]. Neural tube defects (NTDs) are severe congenital birth defects that occur during embryogenesis as a result of environmental and genetic factors, and involve 1‰ of newborns [7]. Open neural tube defects (ONTDs) and closed neural tube defects (CNTDs) are defined based on the affected nerve tissue (exposed or not exposed). ONTDs are common and include open spina bifida, anencephaly, and encephalocele. The clinical manifestations vary depending on the location and severity of the defect [8].

Those affected by trisomy 21 and 18 often exhibit physical and mental retardation, multiple malformations, and fertility disorders, which place a heavy burden on patients, families, and society. There is no effective treatment for trisomy 21 and 18; however, prenatal screening can identify pregnant women at high risk, a prenatal diagnosis can be stablished, and termination of pregnancy is an option. Therefore, it is particularly important to take appropriate screening measures and medical interventions during the first trimester of pregnancy. Based on different screening times, there exist first-trimester screening (FTS), second-trimester screening (STS), and first- and second-trimester combined screening (FSTCS) [2, 9].

A retrospective cohort analysis was performed to collect data from 108,118 pregnant women who underwent FTS, ISTS, and FSTCS between January and December 2019 in Hangzhou, China. In addition, the pregnancy outcomes and results of chromosomal karyotype analysis of amniotic fluid cells were collected to explore the diagnostic value of different screening protocols for prediction of fetal trisomy 21 18 and NTDs, and to determine the optimal screening method to guide clinical prenatal screening.

## 2. Subjects and methods

### 2.1 Study participants

We collected data from 108,118 pregnant women between January and December 2019. The patients were 9–13$^{+6}$ weeks and 15–20$^{+6}$ weeks gestation who were screened in the antenatal screening laboratories of four hospitals in Hangzhou Women's Hospital (Hangzhou Maternity and Child Health Care Hospital), Hangzhou Yuhang District Maternity and Child Health Hospital, Zhejiang Xiaoshan Hospital and Hangzhou Fuyang Woman And Children Hospital. The gravidas were screened as follows: FTS, 72,096; ISTS, 36,022; and FSTCS, 67,631. The flowchart of screening program is shown in Fig 1. Before blood collection, the maternal name, date of birth, gestational age, weight, last menstrual period, cigarette smoking status, history of diabetes and abnormal pregnancies, as well as other information were confirmed. Every pregnant

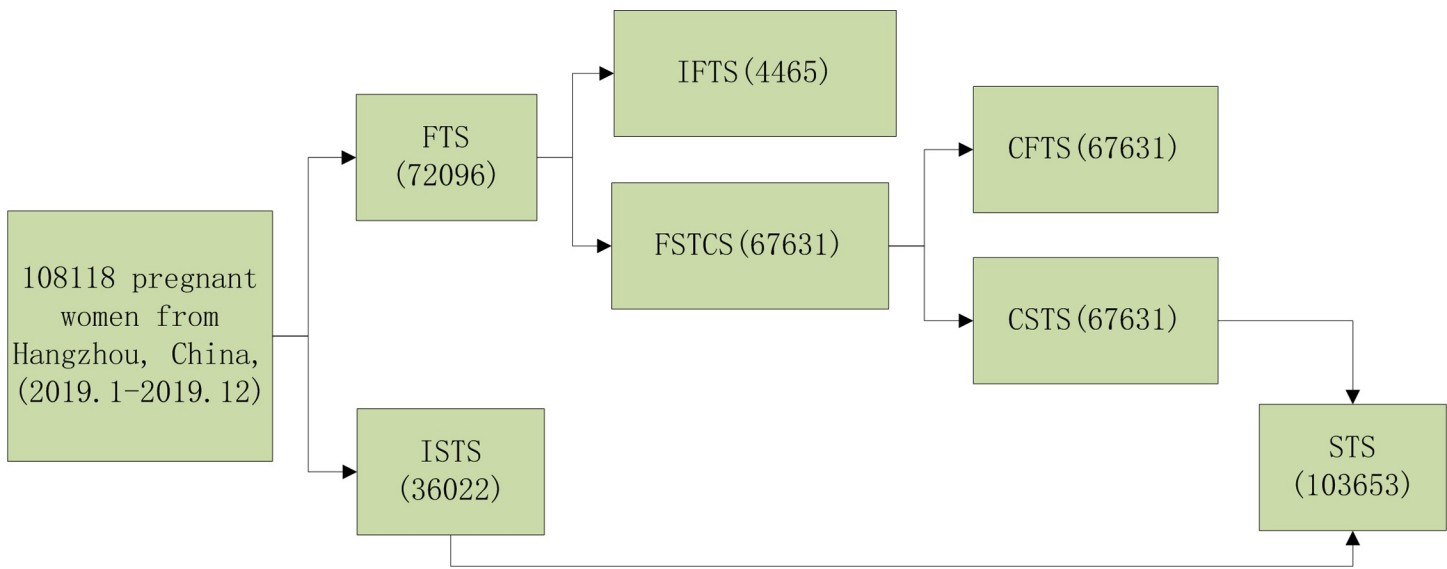

**Fig 1. Prenatal screening program for 108,118 pregnant women.** FTS: First-trimester screening; IFTS: Individual first-trimester screening; CFTS: Combined first-trimester screening; STS: Second-trimester screening; ISTS: Individual second-trimester screening; CSTS: Combined second-trimester screening; FSTCS: first- and second-trimester combined screening.

woman signed an informed consent prior to prenatal screening. This study was approved by the Medical Ethics Committee of the Hangzhou Women's Hospital [2021] Medical Ethics Review A (3) -02.

## 2.2 Screening indicators and methods

FTS was performed to determine the pregnancy-associated plasma protein-A (PAPP-A) and free beta subunit of human chorionic gonadotropin (β-hCG) levels at 9–13$^{+6}$ weeks gestation and/or ultrasound fetal nuchal thickness (NT) at 11–13$^{+6}$ gestation. Patients who had undergone first-trimester screening, but had not been screened in second-trimester screening are referred to as individual first-trimester screening (IFTS). Patients who had undergone first-trimester screening and had undergone second-trimester screening, then participated in the joint screening, are referred to as combined first-trimester screening (CFTS). STS was performed to determine the maternal serum alpha-fetoprotein (AFP) and free β-hCG levels at 15–20$^{+6}$ weeks gestation. Patients who did not participate in first-trimester screening and could not participate in the joint screening in the later stage are referred to as individual second-trimester screening (ISTS). Those who participated in first-trimester screening, followed by second-trimester screening, and participated in joint screening are referred to as combined second trimester screening (CSTS), as shown in Fig 1. FSTCS involved a triple- or quadruple-screening protocol with determination of AFP and free β-hCG levels in the second trimester and matching PAPP-A and/or NT outcomes in the first trimester. The specific FSTCS methodology involved reporting the high risk, but not low risk FTS results, awaiting the STS results, then combining the FTS results to evaluate the probability of a fetus with trisomy 21 or 18.

## 2.3 Reagents and instruments

A 1235 Automatic Time-resolved Fluorescence Immunity System (PerkinElmer, Shelton, CT, USA) was used for detection with PAPP-A and free β -hCG kits, enhancer, washing liquid, quality control samples, and a range of standards (PerkinElmer).

## 2.4 Specimen collection and detection

Eighty-six hospitals in Hangzhou were qualified for blood collection after training. Fasting venous blood (2–3 mL) was collected, and the serum samples were separated by centrifugation at 2500 rpm for 10 min. The samples were stored in a refrigerator at 2–8˚C, then sent to four antenatal screening laboratories for testing by professional cold chain logistics companies within 1 week. The time-resolved fluorescence immunity (DELFIA) method was used and the detection procedures were carried out according to the manufacturer's instructions.

## 2.5 Determination method and screening standard for NT

Fetal neck thickness (NT) was examined at 11–13$^{+6}$ weeks of gestation. Fetal NT was screened according to the standards issued by the Fetal Medicine Foundation (https://fetalmedicine. com/). Specially-trained physicians performed ultrasound examinations according to standardized protocols to assess the fetal NT. In the midsagittal view of the fetus with a natural posture only showing the fetal head and upper chest by magnifying the image, the widest echolucent place between the skin and cervical soft tissue was measured. The fetus was normal when the NT was < 2.5 mm, but was considered abnormal if the NT was ≥ 2.5 mm.

## 2.6 PAPP-A, free β-hCG, and AFP levels, and the NT were represented by multiple of Median (MoM)

$$\text{MoM was defined by the formula}: MoM = \frac{Original\ \ Conj.}{Median} \tag{1}$$

"Original Conj" was the original concentration of PAPP-A, free β -hCG, and AFP, and the NT, and "median" was the median of the original concentration of the corresponding indicators. To reduce the deviation caused by different gestational ages and maternal weight, we used the median equation of gestational age and median equation of maternal weight from the four different hospitals to calibrate the MoM values of various indicators. The MoM value was adjusted according to the median equation, and the adjusted MoM value was used in the risk modeling calculation [10, 11].

$$_{GA\_Med=10}(10.6589 - 0.4597 \times GA + 0.007377 \times GA^2 - 0.000048822 \times GA^3 + 0.0000001165899 \times GA^4) \tag{2}$$

"GA" represented gestational age and" Med" represented median.

$$Weight\_Med = 0.43391 + \frac{37.643}{weight} \tag{3}$$

$$Adjusted\_MoM = \frac{MoM}{GA\_Med \times \text{maternal w}eight\_Med} \tag{4}$$

## 2.7 Risk rate judgment

Lifecycle 4.0 software (PerkinElmer) was used to calculate the risk of trisomy 21 and 18 by combining the maternal age, weight, and gestational age. Cut-off values were defined according to the 2010 Ministry of Health of the People's Republic of China mandatory standards for the industry of prenatal screening by maternal serology in the second trimester, and high risk was defined by trisomy 21 ≥ 1:270, trisomy 18 ≥ 1:350, and AFP MoM ≥ 2.5 [12]. The intermediate risk of trisomy 21 was defined as 1:270–1000 and the risk of trisomy 18 was 1:350–

1000 [13]. Fetal chromosomal karyotype analysis of amniotic fluid cells was recommended in all gravidas of advanced maternal age judged to be at high-risk and low-risk, and the diagnosis was confirmed by ultrasound in those at high risk for a NTD.

## 2.8 Follow-up pregnancy outcomes

Each screened live born was followed in a tertiary network. Karyotype analysis was performed to confirm the diagnosis in spontaneous abortion and intrauterine fetal death. Other abnormalities refer to conditions other than trisomy 21 and 18, and NTD, including fetal trisomy 13 and other chromosome number and structure abnormalities.

## 2.9 Statistical analysis

IBM SPSS 24.0 statistics software (IBM Corp., Armonk, NY, USA) was used for statistical processing. A one sample Kolmogorov-Smirnov test was used to determine if the data were normally distributed. The high risk trisomy 21 and 18 positivity rates were compared by a chi-square test with contingency tables of multiple independent samples. A non-parametric test (Mann-Whitney U test) was used to compare the screening marker MoM level between groups. The data of markers during the second trimester (AFP MoM and free β-hCG MoM) included those who underwent ISTS and CSTS. A $P < 0.05$ was considered statistically significant.

## 3. Results

### 3.1 Comparison of basic demographic data

For FTS, the median maternal age was 28.87 years, the proportion of gravidas with advanced maternal age was 0.55%, the median gestational age was 89 days ($12^{+5}$ weeks), and the median maternal weight was 54 kg. For ISTS, the median maternal age was 29.98 years, the proportion of gravidas with advanced age was 22.80%, the median gestational age was 119 days (17 weeks), and the median maternal weight was 56 kg. For FSTCS, the median maternal age was 28.90 years, the proportion of gravidas with advanced maternal age was 0.49%, the median gestational age was 119 days (17 weeks), and the median maternal weight was 55 kg.

The gestational age confirmation of FTS and FSTCS was mainly based on fetal crown-rump length (CRL) or fetal biparietal diameter (BPD) by ultrasound (CRL in 74.67% and BPD in 74.58% of cases), while ISTS was mainly determined by the last menstrual period (LMP [78.63%]). There were statistically significant differences between the modes of gestational age confirmation among FTS, ISTS and FSTCS ($\chi^2 = 36257.954$, $P < 0.001$). Among 108,118 pregnant women, 3044 cases were diagnosed as abnormal by prenatal diagnosis. Fifty-six cases were detected as abnormal by NTD or second-trimester sonographic markers (lemon and banana signs), and the detection rate of ultrasound was 2.82%. Gravidas with advanced maternal age and those with twins were most frequently tested with ISTS (22.80% and 0.27%, respectively), with statistically significant differences among FTS, ISTS and FSTCS ($\chi^2 = 29415.301$, $P < 0.001$ and $\chi^2 = 216.659$, $P < 0.001$). The method of screening did not differ between gravidas who smoked cigarettes and/or had a history of insulin-dependent diabetes mellitus ($\chi^2 = 0.313$ and $\chi^2 = 1.207$, both $P > 0.05$), as shown in Table 1.

### 3.2 Comparison of high and intermediate risk positivity rates among the different screening methods

The high risk trisomy 21 and 18 positivity rates based on FTS were 2.71% and 0.14%, respectively, while the intermediate risk trisomy 21 and 18 positivity rates based on FTS were 7.19%

and 0.28%, respectively (Table 2). The high risk trisomy 21, trisomy 18, and NTD positivity rates based on ISTS were 9.02%, 0.66%, and 0.52%, respectively, while the intermediate risk trisomy 21 and 18 positivity rates were 16.14% and 1.25%, respectively (Table 2). The high risk trisomy 21, trisomy 18, and NTD positivity rates based on FSTCS were 2.40%, 0.10%, and 0.34%, respectively, while the intermediate risk trisomy 21 and 18 positivity rates were 5.57% and 0.16%, respectively (Table 2). ISTS had the highest high and intermediate risk positivity rates for trisomy 21, trisomy 18, and NTD; the differences were statistically significant among FTS, ISTS and FSTCS (all $P < 0.001$; Table 2).

### 3.3 Comparison of the target disease incidence was based on FTS, ISTS, and FSTCS

The incidence of trisomy 21 in pregnant women screened by FTS, ISTS, and FSTCS was 0.49‰, 0.89‰, and 0.33‰, respectively (Table 3); there were statistically significant differences among three screening methods ($\chi^2 = 6.655$, $\chi^2 = 13.239$, all $P < 0.05$). The incidence of trisomy 18 in pregnant women screened by FTS, ISTS, and FSTCS was 0.13‰, 0.14‰, and 0.09‰, respectively; there were no statistically significant differences among three screening methods ($\chi^2 = 0.009$, $\chi^2 = 0.184$, both $P > 0.05$). The detection rate of other abnormalities was highest in the ISTS group.

### 3.4 Comparison of screening efficiency with FTS, ISTS, and FSTCS

The screening detection rate for trisomy 21 among intermediate risk gravidas by ISTS was higher than FSTCS and FTS; there were no statistically significant differences among the three screening methods ($\chi^2 = 3.028$, $P = 0.220$). The detection rate of trisomy 21 based on ISTS was as follows (Table 4): advanced maternal age > young pregnant women. Pregnant women with

**Table 1. Basic demographic data of pregnant women in three different screening programs n (%).**

| Factors | Types | FTS | ISTS | FSTCS | P |
|---|---|---|---|---|---|
| Gestational age determination method | | | | | $< 0.001^*$ |
| | LMP | 18124 (25.14) | 28323 (78.63) | 17062 (25.23) | |
| | CRL or BPD | 53836 (74.67) | 7383 (20.5) | 50441 (74.58) | |
| | Assisted reproduction | 136 (0.19) | 316 (0.88) | 128 (0.19) | |
| Number of fetus | | | | | $< 0.001^*$ |
| | Singleton | 72071 (99.97) | 35925 (99.73) | 67617 (99.98) | |
| | Twins | 25 (0.03) | 97 (0.27) | 14 (0.02) | |
| Smoking | | | | | 0.855 |
| | No | 71820 (99.62) | 35891 (99.64) | 67382 (99.63) | |
| | Yes | 276 (0.38) | 131 (0.36) | 249 (0.37) | |
| Type I diabetes mellitus | | | | | 0.547 |
| | No | 72028 (99.91) | 35995 (99.93) | 67567 (99.91) | |
| | Yes | 68 (0.09) | 27 (0.07) | 64 (0.09) | |
| Maternal age | | | | | $< 0.001^*$ |
| | < 35 Years | 71697 (99.45) | 27810 (77.2) | 67298 (99.51) | |
| | ≥ 35 Years | 399 (0.55) | 8212 (22.8) | 333 (0.49) | |
| Total | | 72096 | 36022 | 67631 | |

[a]FTS: first-trimester screening; ISTS: individual second-trimester screening; FSTCS: and first and second-trimester combined screening; CRL: crown-rump length; BPD: fetal head biparietal diameter; LMP: last menstrual period.

$^*P < 0.001$.

**Table 2. High and intermediate risk composition ratios n (%).**

| Target disease risk | FTS | ISTS | FSTCS |
|---|---|---|---|
| | n = 72096 | n = 36022 | n = 67631 |
| High risk of Trisomy 21 | 1954 (2.71) | 3249 (9.02) | 1623 (2.40) |
| High risk of Trisomy 18 | 100 (0.14) | 237 (0.66) | 67 (0.10) |
| NTD positive | -- | 188 (0.52) | 233 (0.34) |
| High risk total | 2047 (2.84) | 3639 (10.10) | 1893 (2.80) |
| intermediate risk of Trisomy 21 | 5182 (7.19) | 5815 (16.14) | 3764 (5.57) |
| intermediate risk of Trisomy 18 | 199 (0.28) | 450 (1.25) | 105 (0.16) |
| intermediate risk total | 5272 (7.31) | 6115 (16.98) | 3812 (5.64) |

[a]FTS: first-trimester screening; ISTS: individual second-trimester screening; FSTCS: and first and second-trimester combined screening; NTD: neural tube defects.

NT detection in FTS < non-NT detection. The detection rate for trisomy 18 based on FSTCS was higher, followed by FTS and ISTS; there were no statistically significant differences among the three screening methods ($\chi^2$ = 0.072, $P$ = 0.964). FTS had the highest PPV for trisomy 21 (0.87%) and 18 (6.00%). FSTCS had the lowest false-positive rate for trisomy 21 and 18 (2.34% and 0.09%, respectively).

## 3.5 Comparison of MoM values based on FTS, STS, and FSTCS

PAPP-A and AFP MoM values of pregnant women with trisomy 21 fetuses were decreased (Fig 2A and 2D); the differences between the trisomy 21 and Non-Trisomy 21 fetuses groups were statistically significant (all $P$ < 0.001), as shown in Table 5. The free β-hCG and NT MoM levels were both increased (Table 5 and Fig 2B, 2C and 2E); the difference was statistically significant between the trisomy 21 and Non-Trisomy 21 fetuses groups ($P$ < 0.001). The PAPP-A, free β-hCG, and AFP MoM levels of pregnant women with trisomy 18 fetuses were all decreased (Fig 2F, 2G, 2I and 2J); the differences between the trisomy18 and Non-Trisomy 18 fetuses groups were statistically significant (all $P$ < 0.001), as shown in Table 6. Pregnant women with trisomy 18 fetuses had an increased NT MoM values (Table 6 and Fig 2H; 1.18 95%CI (1.05–1.95), $P$ = 0.010). The free β-hCG MoM level was decreased and the AFP MoM level was increased in pregnant women with NTD fetuses (Fig 2K and 2L).

## 4. Discussion

This was a retrospective control study involving 108,118 pregnant women who received three types of prenatal screening tests during the first (9–13$^{+6}$ weeks) and second trimesters (15–20$^{+6}$ weeks) in Hangzhou. We evaluated the efficacy of three different maternal screening programs (FTS, ISTS, and FSTCS) in predicting offspring with trisomy 21, trisomy 18, and neural tube defects (NTDs). ISTS for trisomy 21 screening had the highest detection rate, followed by FSTCS and FTS. FTS and FSTCS had the highest detection rates for trisomy 18. There were no

**Table 3. Confirmed sample distribution and incidence statistics n (‰).**

| Screening program | n | Trisomy 21 | Trisomy 18 | NTD | Other anomalies |
|---|---|---|---|---|---|
| FTS | 71977 | 35 (0.49) | 9 (0.13) | 10 (0.14) | 65 (0.90) |
| ISTS | 35936 | 32 (0.89) | 5 (0.14) | 3 (0.08) | 58 (1.61) |
| FSTCS | 67536 | 22 (0.33) | 6 (0.09) | 8 (0.12) | 59 (0.87) |

[a]FTS: first-trimester screening; ISTS: individual second-trimester screening; FSTCS: and first and second-trimester combined screening; NTD: neural tube defects.

**Table 4. Comparison of screening efficiency among different screening programs.**

| groups | n | Confirmed cases (n) | incidence ‰ | Confirmed cases results | | | PR (%) | FPR (%) | DR* (%) | PPV (%) | DR# (%) |
|---|---|---|---|---|---|---|---|---|---|---|---|
| | | | | High risk (n) | intermediate risk (n) | Low risk (n) | | | | | |
| Trisomy 21 | | | | | | | | | | | |
| FTS | 72096 | 35 | 0.49 | 17 | 9 | 9 | 2.71 | 2.69 | 48.57 | 0.87 | 74.29 |
| Non-NT | 25703 | 11 | 0.43 | 6 | 3 | 2 | 4.97 | 4.95 | 54.55 | 0.47 | 81.82 |
| NT | 46393 | 24 | 0.52 | 11 | 6 | 7 | 1.46 | 1.43 | 45.83 | 1.63 | 70.83 |
| ISTS | 36022 | 32 | 0.89 | 22 | 7 | 3 | 9.02 | 8.97 | 68.75 | 0.68 | 90.63 |
| AMA (≥35 years) | 8212 | 22 | 2.68 | 17 | 3 | 2 | 20.98 | 20.83 | 77.27 | 0.99 | 90.91 |
| Non-AMA | 27810 | 10 | 0.36 | 5 | 4 | 1 | 5.49 | 5.47 | 50.00 | 0.33 | 90.00 |
| FSTCS | 67631 | 22 | 0.33 | 14 | 3 | 5 | 2.40 | 2.34 | 63.64 | 0.86 | 77.27 |
| Trisomy 18 | | | | | | | | | | | |
| FTS | 72096 | 9 | 0.12 | 6 | 0 | 3 | 0.14 | 0.13 | 66.67 | 6.00 | 66.67 |
| Non-NT | 25703 | 2 | 0.08 | 1 | 0 | 1 | 0.30 | 0.30 | 50.00 | 1.30 | 50.00 |
| NT | 46393 | 7 | 0.15 | 5 | 0 | 2 | 0.05 | 0.04 | 71.43 | 21.74 | 71.43 |
| ISTS | 36022 | 5 | 0.14 | 3 | 0 | 2 | 0.66 | 0.65 | 60.00 | 1.27 | 60.00 |
| AMA (≥ 35 years) | 8212 | 4 | 0.49 | 3 | 0 | 1 | 1.58 | 1.55 | 75.00 | 2.31 | 75.00 |
| Non-AMA | 27810 | 1 | 0.04 | 0 | 0 | 1 | 0.38 | 0.38 | 0 | 0 | 0 |
| FSTCS | 67631 | 6 | 0.09 | 4 | 1 | 1 | 0.10 | 0.09 | 66.67 | 5.97 | 83.33 |

[a]FTS: first-trimester screening; ISTS: individual second-trimester screening; FSTCS: and first and second-trimester combined screening; NT: nuchal thickness; AMA: Advanced maternal age (≥ 35 years); PR, positive rate; FPR, false positive rate; DR; detection rate without intermediate risk; PPV, positive predictive value; DR# detection rate with intermediate risk and high risk.

statistical differences in the screening detection rates for trisomy 21 and 18 among the three screening programs (all $P > 0.05$). We concluded that FSTCS was superior to FTS and ISTS screening and substantially reduced the number of high risk pregnancies for trisomy 21 and 18; however, FSTCS was not significantly different in detecting fetal trisomy 21 and 18 and other confirmed cases with chromosomal abnormalities.

With advances in prenatal research, serologic prenatal screening has also advanced. STS was the first screening test to be implemented, and included AFP, free β-hCG, and unconjugated estriol (uE3) levels; FTS (PAPP-A, free β-hCG, and NT) and FSTCS (PAPP-A + AFP + free β-hCG and AFP + free β-hCG + PAPP-A + NT) were subsequently introduced [14, 15]. Studies have shown that the false-positive rate of serum screening alone in early pregnancy was high. When combined with ultrasound markers, such as NT, nasal bone, and blood flow examination of the tricuspid valve and ductus venosus, the false-positive rate can be further reduced and the disease detection rate can be improved [16]. The FTS method in this study was performed as follows: pregnant women without NT results were screened based on serum screening in early pregnancy; and with NT results of serum screening combined with NT. For the FSTCS method, high risk FTS results were reported, while low risk results were not. When the STS results (AFP and free β-hCG) were available, the STS results were combined with the FTS results (PAPP-A and/or NT) [17]. Non-invasive prenatal testing (NIPT) is recommended for pregnant women at intermediate risk [18]. High risk pregnant women are recommended to have chorionic villus or amniotic fluid cell chromosome examination, and low risk pregnant women should be followed up until delivery [12].

The high risk FTS, ISTS, and FSTCS for trisomy 21 positivity rates were 2.71%, 9.02%, and 2.40%, respectively. The high risk FSTCS positivity rate was the lowest, while the high risk

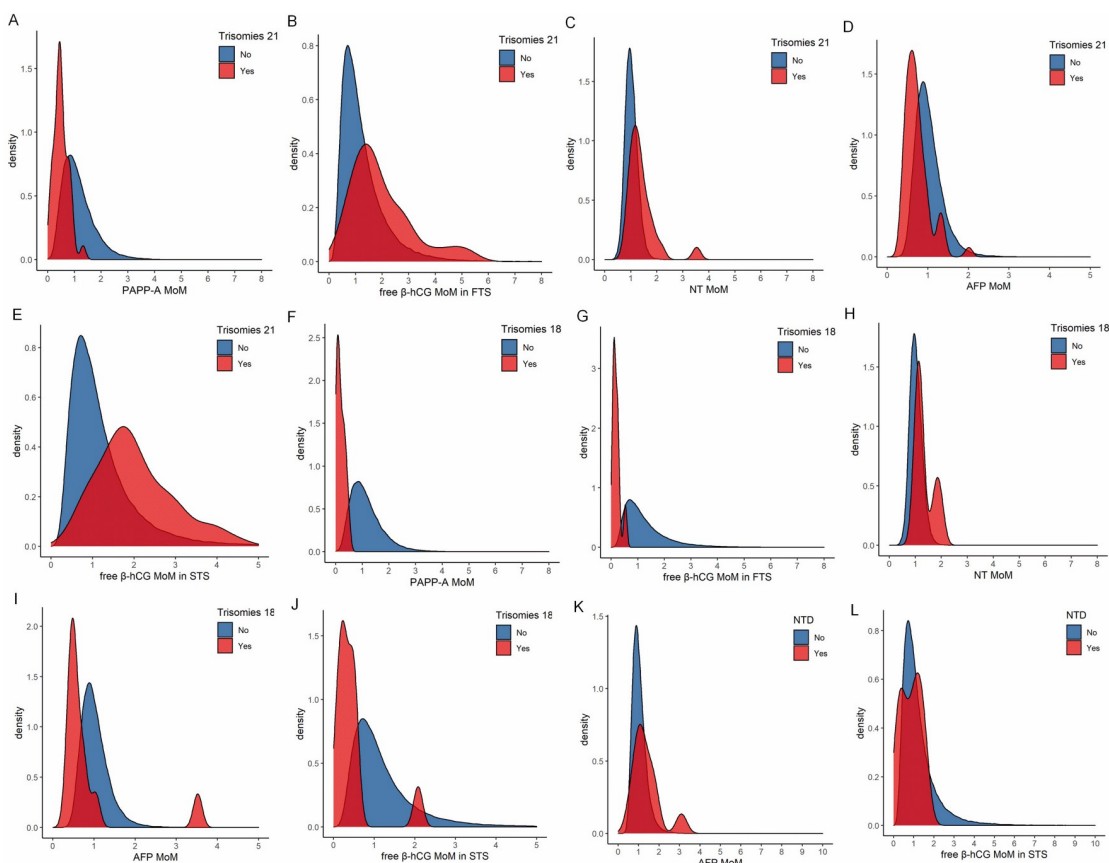

**Fig 2. Probability density diagram of each index for trisomy 21, trisomy 18, and NTDs.**

ISTS positivity rate was the highest; the differences were statistically significant among three screening methods (all $P < 0.05$). This finding was similar to the corresponding high risk screening positivity rates of 3.44%, 6.11%, and 2.91% in our previous study [19]. Zhang et al. [20] reported that the high risk FSTCS positivity rate was 1.89%, which was 2.93% lower than the second trimester screening, and the detection rate increased by 2.1%. Zhang et al. [20] concluded that the FSTCS method was better than STS, which was similar to the results of the current study. FSTCS had the lowest false-positive rate for trisomy 21 and 18 (2.34% and 0.09%,

**Table 5. Distribution of marker MoM values in the serum of pregnant women with or without trisomy 21 fetuses.**

| markers | Non-Trisomy 21 | | Trisomy 21 | | *P* |
|---|---|---|---|---|---|
| | n | MoM (95%CI) | n | MoM (95%CI) | |
| PAPP-A | 72061 | 1.02 (0.40–2.25) | 35 | 0.46 (0.10–0.94) | < 0.001* |
| free β-hCG in FTS | 72061 | 1.03 (0.41–2.88) | 35 | 1.68 (0.67–5.26) | < 0.001* |
| NT | 46369 | 1.00 (0.67–1.43) | 24 | 1.29 (0.87–2.18) | < 0.001* |
| AFP | 103599 | 0.97 (0.59–1.65) | 54 | 0.68 (0.40–1.36) | < 0.001* |
| free β-hCG in STS | 103599 | 0.99 (0.40–2.87) | 54 | 2.00 (0.80–7.48) | < 0.001* |

[a]PAPP-A: pregnancy-associated plasma protein A; free β-hCG: free beta-subunit of human chorionic gonadotropin; NT: nuchal transparency; AFP: alpha-fetoprotein; MoM: multiple of the median. FTS: first-trimester screening; STS: second-trimester screening; Data are presented as median ($P_{2.5}$–$P_{97.5}$).
*$P < 0.001$.

**Table 6. Distribution of marker MoM values in the serum of pregnant women with or without trisomy 18 fetuses.**

| markers | Non-Trisomy 18 | | Trisomy 18 | | P |
|---|---|---|---|---|---|
| | n | MoM (95%CI) | n | MoM (95%CI) | |
| PAPP-A | 72087 | 1.02 (0.40–2.25) | 9 | 0.11 (0.02–0.45) | < 0.001* |
| free β-hCG in FTS | 72087 | 1.03 (0.41–2.88) | 9 | 0.15 (0.07–0.53) | < 0.001* |
| NT | 46386 | 1.00 (0.67–1.43) | 7 | 1.18 (1.05–1.95) | 0.010** |
| AFP | 103642 | 0.97 (0.59–1.65) | 11 | 0.52 (0.39–3.52) | < 0.001* |
| free β-hCG in STS | 103642 | 0.99 (0.40–2.88) | 11 | 0.28 (0.07–2.08) | < 0.001* |

[a]PAPP-A: pregnancy-associated plasma protein A; free β-hCG: free beta-subunit of human chorionic gonadotropin; NT: nuchal transparency; AFP: alpha-fetoprotein; MoM: multiple of the median. FTS: first-trimester screening; STS: second-trimester screening; Data are presented as median ($P_{2.5}$–$P_{97.5}$).

*$P < 0.001$;

**$P < 0.05$.

respectively), thus showing that the FSTCS method reduced the high risk positive rate, reduced the number of amniocenteses and NIPTs, and reduced the psychological and economic burden among pregnant women.

This study showed that the FTS, ISTS, and FSTCS detection rates for trisomy 21 were 48.57%, 68.75%, and 63.64%, respectively. The ISTS method had the highest detection rate for trisomy 21, which was different from the Zheng study [21], in which the FSTCS detection rate was higher than FTS and STS, and the Wright study [22] in which the false-positive rate of the FTS method for trisomy 21 was 3%–5% and the detection rate was 90%–95%. This finding may reflect the tendency for gravidas of advanced maternal age in this region to participate more in STS and less in FTS, which may also be caused by the large sample size of this study. In addition, the FTS, ISTS, and FSTCS detection rates for trisomy 18 were 66.67%, 60.00%, and 66.67%, respectively, in the current study. Palomaki et al. [23] used a combination of serum markers (PAPP-A in early pregnancy and AFP, uE3, and free β-hCG in second pregnancy) to screen for trisomy 18 and reported a detection rate of 90% when the false-positive rate was 0.1%, which was reduced to 67% when PAPP-A in early pregnancy was removed from the analysis. Our results differed may be related to the patient demographics, sample size, and screening indicators across laboratories.

With respect to trisomy 21 screening, the current study suggested that the intermediate risk detection rates with FTS, ISTS, and FSTCS were 25.72%, 21.88%, and 13.63%, respectively, higher than patients without intermediate risk. For trisomy 18, FSTCS increased by 16.66%, and FTS and ISTS did not change. The results suggested that combined intermediate risk may be increased in nearly 25% of pregnant women to perform prenatal diagnosis to reduce missed tests. Luo et al. [13] showed that combined first trimester screening (CFTS) as first-line screening had the lowest cost and higher detection rate (93.94%) when the intermediate risk was 1:51–1500 was compared with other screening programs. Therefore, according to local health and economic status, an appropriate screening program and cut-off values can be adopted as first-line screening, which will help to establish a better cost-effective screening model. Younesi et al. [24] showed that intermediate risk is important because 23 of 45 false-negative results were in the risk range 1:250–1100. The free β-hCG MoM, PAPP-A MoM, and NIPT had abnormal results in eight of 23 false-negative cases, which confirmed the benefit of setting an intermediate risk.

NTD is a serious neurological defect of the fetus, causing a great challenge to the families and societies [25]. Currently, the diagnosis of ONTD mainly relies on ultrasound images and serum AFP in the second trimester in the maternal serum [25]. Our study showed that the detection

rate of NTD in FTS were higher than in ISTS, which may be because that NT examination improved the detection rate of FTS. Therefore, NTD screening should be combined with ultrasound images and maternal serum screening. In addition, Our preliminary research Showed that AFP variants (AFP-L2 and AFP-L3) are favorable biomarkers for screening ONTD, and it have superior sensitivity and specificity [26]. Gerardo et al. [27] showed that abnormal intracranial translucency during the first-trimester may be a useful screening marker for early detection of NTDs. Studies have shown that advanced maternal age is associated with adverse pregnancy outcomes, such as preterm birth, miscarriage, and fetal chromosome abnormalities [28]. In this study, we found that the incidence of trisomy 21 and trisomy 18 was higher in advanced maternal age than in non-advanced maternal age. The detection rate for fetal chromosomal malformations of advanced maternal age in ISTS was higher than that of non-advanced maternal age. But studies have showed that the effect of advanced maternal age on fetal chromosome abnormalities has been overestimated, introducing absolute risk (eliminating the mother's age risk) from the risk algorithm may reduce the screening positive rate for advanced maternal age and increase the screening positive rate for non-advanced maternal age [29].

Recently, NIPT using maternal plasma fetal-free DNA or placenta-specific mRNA, has gradually been applied to prenatal screening for chromosomal abnormalities [30]. Moreover, the high sensitivity and specificity of NIPT have been confirmed [31]. NIPT is a highly effective screening program for chromosomal abnormalities. To reduce the number of invasive prenatal diagnoses, some high and intermediate risk pregnant women may be recommended to undergo NIPT screening first, and amniocentesis may be used for karyotyping if the result is positive. The cost of NIPT is relatively high, thus NIPT and serologic prenatal screening should be used in combination with the conditions of the pregnant woman to reduce the missed detection of fetuses with chromosomal abnormalities. Undeniably, NIPT, as a non-invasive, highly sensitive and specific feature, has become a novel screening method for fetal chromosome malformations. With the rapid development of molecular biology, perhaps one day in the future, the traditional maternal serum biochemical screening method may be a secondary screening method.

The basis of serum prenatal screening is that pregnant placentas will produce PAPP-A, AFP, free β-hCG, and other substances, when the fetus is affected (chromosomal abnormalities and neural tube malformations). Indeed, these substances can be discharged into the amniotic fluid of pregnant women through the urine of the fetus, then transported to the blood of pregnant women. Therefore, PAPP-A, AFP, and free-hCG can be used for screening fetal diseases. PAPP-A is a glycoprotein secreted by placental trophoblast cells, which is closely related to trisomy 21 and other chromosomal abnormalities [32]. Wald et al. [33] reported that PAPP-A was decreased by 60% in the serum of pregnant women with a trisomy 21 fetus, which is consistent with the results of the current study. AFP is a fetal-derived glycoprotein with a molecular weight of 69 KD that is mainly produced by the yolk sac and fetal liver; AFP can be detected in the first trimester [34]. Relevant studies have shown that AFP in the serum of pregnant women with trisomy 21 or 18 fetuses was lower than the serum of healthy fetuses of pregnant women, and AFP MoM with trisomy 18 fetuses was even lower [35]. Therefore, we must be vigilant and extend extra attention to cases with abnormally elevated or decreased screening indicators.

The limitations of this study were as follows: First, the cohort did not use a unified combined screening protocol for first and second trimesters. One-third of pregnant women were missing FTS. A small number of pregnant women who participated in FTS did not take the second trimester test, but directly entered the prenatal diagnosis. These factors can affect the integrity of screening program data. Second, there were only three cases of trisomy 13 in this cohort, so we did not list trisomy 13 separately, but included it among the other abnormalities. In addition a small percentage of patients with twin gestations voluntarily opted for serologic

screening. Therefore, we included this part of the data of women with twin gestations in the study. Finally, due to some policies related to prenatal screening in China, older pregnant women are usually advised to undergo direct mid-trimester screening. Therefore, older pregnant women in the region tend to be more involved in ISTS, which is one of the differences between this cohort and others in other regions.

## 5. Conclusions

In conclusion, FSTCS was superior to FTS and ISTS screening and substantially reduced the number of high risk pregnancies for trisomy 21 and 18. FSTCS reduced the high risk positive rate of chromosomal abnormal diseases, reduced the number of villus and amniotic fluid cell examinations in pregnant women, and thus reduced the psychological pressure and economic burden in pregnant women. Although compared to ISTS, FSTCS did not significantly improve the rate of chromosome disease detection. However, for eligible pregnant women, we still recommend FSTCS, because it will reduce the high risk of chromosomal diseases and reduce the probability of invasive diagnosis in pregnant women.

## Supporting information

**S1 Checklist. STROBE statement—checklist of items that should be included in reports of observational studies.**
(DOCX)

**S1 Data.**
(XLSX)

**S1 File.**
(PDF)

**S2 File.**
(DOC)

## Acknowledgments

We gratefully express our gratitude to some teachers for their support and help in this experiment, including Xuelian Chu and Linyuan Gu from the antenatal screening laboratory of Yuhang District Maternal and Child Health Hospital in Hangzhou, Jun Liu and Liufen Gu from the antenatal screening laboratory of Zhejiang Xiaoshan Hospital, and Haiya He from the antenatal screening laboratory of Hangzhou Fuyang Woman And Children Hospital. We thank International Science Editing (http://www.internationalscienceediting.com) for editing this manuscript.

## Declaration

It was conformed the Enhancing the QUAlity and Transparency Of health Research (EQUATOR) network guidelines.

## Author Contributions

**Data curation:** Yezhen Shi, Liyao Li.

**Methodology:** Xiaoying Wang.

**Writing – original draft:** Yiming Chen, Wenwen Ning.

**Writing – review & editing:** Yijie Chen, Wen Zhang.

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
