## [Decision Letter · Decision Letter 0]

24 Aug 2022

PONE-D-22-00752Maternal prenatal screening programs that predict trisomy 21, trisomy 18, and neural tube defects in offspringPLOS ONE

Dear Dr. Chen,

Thank you for submitting your manuscript to PLOS ONE. After careful consideration, we feel that it has merit but does not fully meet PLOS ONE’s publication criteria as it currently stands. Therefore, we invite you to submit a revised version of the manuscript that addresses the points raised during the review process.

Please note that we have only been able to secure a single reviewer to assess your manuscript. We are issuing a decision on your manuscript at this point to prevent further delays in the evaluation of your manuscript. Please be aware that the editor who handles your revised manuscript might find it necessary to invite additional reviewers to assess this work once the revised manuscript is submitted. However, we will aim to proceed on the basis of this single review if possible.

The reviewers have raised a number of major concerns. They request improvements to the reporting of methodological aspects of the study. The reviewers also note concerns about the statistical analyses presented and request re-analyses be completed.

Could you please carefully revise the manuscript to address all comments raised?

We look forward to receiving your revised manuscript.

Kind regards,

Thomas Phillips, PhD

Staff Editor

PLOS ONE

Journal Requirements:

“This study was supported by HangZhou Medicine and Health Science and Technology Research Project (Z20220025)and Yiming Chen was chief investigator.”

Reviewers' comments:

Reviewer's Responses to Questions

**Comments to the Author**

1. Is the manuscript technically sound, and do the data support the conclusions?

Reviewer #1: Partly

2. Has the statistical analysis been performed appropriately and rigorously? 

Reviewer #1: I Don't Know

3. Have the authors made all data underlying the findings in their manuscript fully available?

Reviewer #1: Yes

4. Is the manuscript presented in an intelligible fashion and written in standard English?

Reviewer #1: No

5. Review Comments to the Author

Reviewer #1: Thank you for your submission. It is good to have more data on the performance of prenatal screening programs for fetal diseases in China.

There is an extensive body of evidence in the medical literature that reports and analyzes the performance of the different screening modalities for aneuploidy in the first and second trimesters of pregnancy.

STUDY DESIGN:

The research subject is relevant, and the sample size is very good.

However, some issues stand out regarding study design:

- The authors decided to analyze the screening performance for both aneuploidy and neural tube defects together. Most of the published studies have analyzed these results separately.

- Another curious design decision was to include twin pregnancies together with singleton pregnancies when it is well known that the screening performance differs significantly in both groups. Moreover, even within twins, some factors can influence significantly the detection rate of aneuploidy and structural anomalies, such as chorionicity.

- for some reason, the performance of screening for trisomy 13 was not included in the analysis

- patients who only had serum biochemistry measurements in the first trimester were included in the analysis together with those who had combined screening. "Double screening" or biochemical screening in the first trimesters is currently considered suboptimal.

READABILITY:

Unfortunately, the manuscript lacks clarity, and the choice of words is not appropriate in many cases, which makes it difficult to follow. There are serious grammatical errors, including unfinished sentences (page 4, line 89) and typos (page 17, line 301, the author's name is “Palomaki”; line ). I strongly recommend that the authors should consider rewriting it with the advice of a native English speaker.

METHODS:

There are also serious conceptual and methodological errors, such as saying that the nuchal translucency was evaluated following the FMF recommendations but then stating that NT thickness was measured "between 15 and 20 weeks” and that “the fetus was normal when NT was 2.5mm”.

The strategy used to screen for neural tube defects is unclear, since, for example, data on ultrasound findings in the first and second trimesters are not included (was intracranial translucency measured in all cases? In what percentage of the cases with FTS was an NTD detected or suspected sonographically? Were second-trimester sonographic markers (lemon and banana signs) evaluated in all cases? What was the sonographic detection rate?)

RESULTS:

Some of the reported results are probably erroneous and require revision because they are not in line with what is known from the literature.

Examples:

page 8, line 166: should say “the median maternal age”

page 8, line 167: “the proportion of gravidas with advanced maternal age was 0.55% “. In most populations, advanced maternal age (defined as maternal age of 35 o more at delivery) ranges between 30-60%, so this sounds strange.

Errors like these undermine the credibility of the results of the analysis.

The tables are not clear and irrelevant data such as the value of the chi-square statistic are reported. For a better understanding, they should be simplified.

DISCUSSION:

The authors reach conflicting conclusions with all the previous experience in aneuploidy screening, such as stating that the biochemical screening of the second trimester had a higher detection rate than the screening of the first trimester. This, together with the low detection rates achieved in the FTS, suggest that the latter was performed with an insufficient level of quality since in many validation studies, the detection rate of the combined screening ranges between 75-90%.

The authors state that “The result shoed that STS for trisomy 21 had the highest detection rats, followed by FSTCS and FTS” but then say that “there were no statistical differences in the detection rates of trisomy 21 among the 3 screening programs”, and later say that “we concluded that FSTCS was superior to FTS and STS screening”. These statements appear to be contradictory.

Rather than provide plausible explanations, the authors suggest that this could be due to " the tendency for gravidas of advanced maternal age in this region to participate more in STS and less in FTS” or issues related to sample size, demographic factors, or “screening indicators across laboratories”

CONCLUSION:

Based on the above, I consider that the manuscript cannot be accepted for publication in its current version and could only be considered in case of a major revision.

6. PLOS authors have the option to publish the peer review history of their article (what does this mean?). If published, this will include your full peer review and any attached files.

Reviewer #1: No

---

## [Author Response · Author response to Decision Letter 0]

25 Oct 2022

Dr. Editor

PLOS ONE 

October 24, 2022

Dear editor

We would like to submit the enclosed manuscript PONE-D-22-00752 entitled “Maternal prenatal screening programs that predict trisomy 21, trisomy 18, and neural tube defects in offspring”, which we wish to be considered for publication in PLOS ONE. We have made changes in response to journal requirements and reviewers' comments, and marked in red font, specific modifications are as follows. Our manuscript has also been linguistically twice retouched by International Science Editing and is presented in an intelligible fashion and in standard English.

Journal Requirements:

Q1. Please ensure that your manuscript meets PLOS ONE's style requirements, including those for file naming.

Answer1: We have revised the manuscript to meet PLOS ONE's style requirements.

Q2. Please include your amended Funding Statement within your cover letter. We will change the online submission form on your behalf.

Answer2: Because the original fund has not been formally issued, we modify the fund as follows: “This study was supported by Hangzhou Medicine and Health Science and Technology Research Project (2017A055) and Yiming Chen was chief investigator. There was no additional external funding received for this study”.

Q3. Your ethics statement should only appear in the Methods section of your manuscript. If your ethics statement is written in any section besides the Methods, please delete it from any other section.

Answer3: We have removed the ethics statement except for the Methods section.

Q4. Please include captions for your Supporting Information files at the end of your manuscript, and update any in-text citations to match accordingly.

Answer4: We have updated the Supporting Information files and in-text citations.

Review Comments to the Author:

Reviewer #1: 

STUDY DESIGN: The research subject is relevant, and the sample size is very good. However, some issues stand out regarding study design:

Q1: The authors decided to analyze the screening performance for both aneuploidy and neural tube defects together. Most of the published studies have analyzed these results separately.

Answer1: Although most published studies have analyzed the screening performance of aneuploidy and neural tube defects separately for these outcomes. However, there is little relevant literature from Hangzhou, China.

Q2: Another curious design decision was to include twin pregnancies together with singleton pregnancies when it is well known that the screening performance differs significantly in both groups. Moreover, even within twins, some factors can influence significantly the detection rate of aneuploidy and structural anomalies, such as chorionicity. 

Answer2: For twin pregnancies we usually recommend NT or other prenatal screening or prenatal diagnosis, but there is always a small percentage of twin pregnancies that voluntarily opt for serological screening, so we also include twin pregnancies together with singleton pregnancies in our analysis.

Q3: for some reason, the performance of screening for trisomy 13 was not included in the analysis

Answer3: Since only 3 cases of trisomy 13 were detected in this cohort, we did not list trisomy 13 separately, but included it among the other abnormalities.

Q4: patients who only had serum biochemistry measurements in the first trimester were included in the analysis together with those who had combined screening. "Double screening" or biochemical screening in the first trimesters is currently considered suboptimal. 

Answer4: Although "Double screening" or biochemical screening in the first trimesters is currently considered suboptimal, some successes have been achieved since the mid-pregnancy duplex screening program was included in the local financial support for the prevention and control of major birth defects in Hangzhou, China.

READABILITY:

Q5: Unfortunately, the manuscript lacks clarity, and the choice of words is not appropriate in many cases, which makes it difficult to follow. There are serious grammatical errors, including unfinished sentences (page 4, line 89) and typos (page 17, line 301, the author's name is “Palomaki”; line). I strongly recommend that the authors should consider rewriting it with the advice of a native English speaker. 

Answer5: Thanks to the reviewers and responsible editors for your corrections. The language and grammar of this manuscript have been revised by the relevant English professionals.

METHODS:

Q6: There are also serious conceptual and methodological errors, such as saying that the nuchal translucency was evaluated following the FMF recommendations but then stating that NT thickness was measured "between 15 and 20 weeks” and that “the fetus was normal when NT was 2.5mm”. 

Answer6: Thanks to the reviewer's correction. We wrote wrongly at that time, and it should be 11-13+6 weeks for NT thickness check, "the fetus is normal when NT is <2.5 mm". 

Q7: The strategy used to screen for neural tube defects is unclear, since, for example, data on ultrasound findings in the first and second trimesters are not included (was intracranial translucency measured in all cases? In what percentage of the cases with FTS was an NTD detected or suspected sonographically? Were second-trimester sonographic markers (lemon and banana signs) evaluated in all cases? What was the sonographic detection rate?)

Answer7: Among 108118 pregnant women, only a small proportion of those with a high risk of NTD by prenatal screening or low risk of NTD were evaluated for mid-pregnancy ultrasound markers (lemon and banana signs).

There were 3044 cases of confirmed abnormal findings after prenatal diagnosis and ultrasound, of which 56 cases were NTD. The ultrasound detection rate was 2.82%.

RESULTS:

Q8: Some of the reported results are probably erroneous and require revision because they are not in line with what is known from the literature. Examples: page 8, line 166: should say “the median maternal age”

Answer8: Thanks to the reviewers and responsible editors for their corrections. We have revised page 8, line 166 to "the median maternal age".

Q9: page 8, line 167: “the proportion of gravidas with advanced maternal age was 0.55% “. In most populations, advanced maternal age (defined as maternal age of 35 o more at delivery) ranges between 30-60%, so this sounds strange. 

Answer9: Since advanced maternal age is not recommended for FTS in early pregnancy in our region, so the advanced maternal age was 0.55%, while ISTCS advanced maternal age was 22.80%, so the participation in FTS and FSTCS is basically for low age pregnant women. The advanced maternal age in ISTS was 22.80%. The above are the difference between our study and others.

Q10: The tables are not clear and irrelevant data such as the value of the chi-square statistic are reported. For a better understanding, they should be simplified. 

Answer10: We modified and simplified the tables accordingly, e.g. deleting the columns of chi-square statistics in Table 1 and the columns of Z statistics in Tables 5-6.

DISCUSSION: 

Q11: The authors reach conflicting conclusions with all the previous experience in aneuploidy screening, such as stating that the biochemical screening of the second trimester had a higher detection rate than the screening of the first trimester. This, together with the low detection rates achieved in the FTS, suggest that the latter was performed with an insufficient level of quality since in many validation studies, the detection rate of the combined screening ranges between 75-90%.

Answer11: In our large retrospective cohort study, the detection rate of biochemical screening for midterm pregnancy alone was higher than that of early pregnancy screening, whereas the detection rate of biochemical screening for midterm pregnancy alone was lower than that of early pregnancy screening in low-risk pregnant women. As for the low detection rate of FTS, it may be related to statistical bias because nearly 1/3 of the pregnant women in the study missed early pregnancy screening.

Q12: The authors state that “The result shoed that STS for trisomy 21 had the highest detection rats, followed by FSTCS and FTS” but then say that “there were no statistical differences in the detection rates of trisomy 21 among the 3 screening programs”, and later say that “we concluded that FSTCS was superior to FTS and STS screening”. These statements appear to be contradictory. 

Answer12: The results of this study showed that ISTS had the highest detection rate for trisomy 21 in 22.80% of high-risk pregnancies, followed by FSTCS and FTS". In conclusion, FSTCS was superior to FTS and ISTS screening and greatly reduced the number of high-risk pregnancies for trisomy 21 and 18. FSTCS reduced the high-risk positive rate for chromosomal abnormal disorders. It is possible that the reviewers did not understand that we mainly emphasize that there is a difference between the different screening protocols for screening high-risk positive rates for the previous, but no difference for the final confirmation of trisomy 21 and 18. STS here refers to ISTS. See Figure 1.

Q13: Rather than provide plausible explanations, the authors suggest that this could be due to " the tendency for gravidas of advanced maternal age in this region to participate more in STS and less in FTS” or issues related to sample size, demographic factors, or “screening indicators across laboratories” 

Answer13: Due to some policies related to prenatal screening in China, older pregnant women are usually advised to undergo direct mid-trimester screening. Therefore, older pregnant women in the region tend to be more involved in ISTS. STS here refers to ISTS. See Figure 1

We confirmed that this manuscript has not been published elsewhere and is not under consideration by another journal. All authors have approved the manuscript and agree with submission to PLOS ONE. The authors have no conflicts of interest to declare.

We shall look forward to hearing from you at your earliest convenience.

* Corresponding author: 

Yiming Chen 

Department of Prenatal Diagnosis and Screening Center Hangzhou Women’s Hospital (Hangzhou Maternity and Child Health Care Hospital), No. 369, Kunpeng Road, Shangcheng District Hangzhou, Zhejiang 310008, China

TEL: +86 56005843

E-mail: cxy40344@163.com

Yours sincerely.

Yiming Chen

---

## [Decision Letter · Decision Letter 1]

6 Dec 2022

PONE-D-22-00752R1Maternal prenatal screening programs that predict trisomy 21, trisomy 18, and neural tube defects in offspringPLOS ONE

Dear Dr. Chen,

Thank you for submitting your manuscript to PLOS ONE. After careful consideration, we feel that it has merit but does not fully meet PLOS ONE’s publication criteria as it currently stands. Therefore, we invite you to submit a revised version of the manuscript that addresses the points raised during the review process.

We look forward to receiving your revised manuscript.

Kind regards,

Antonio Simone Laganà, M.D., Ph.D.

Academic Editor

PLOS ONE

Journal Requirements:

Additional Editor Comments:

The reviewers expressed additional minor concerns: for this reason, I invite the authors to perform the additional changes, as suggested.

Reviewers' comments:

Reviewer's Responses to Questions

**Comments to the Author**

1. If the authors have adequately addressed your comments raised in a previous round of review and you feel that this manuscript is now acceptable for publication, you may indicate that here to bypass the “Comments to the Author” section, enter your conflict of interest statement in the “Confidential to Editor” section, and submit your "Accept" recommendation.

Reviewer #1: All comments have been addressed

Reviewer #2: (No Response)

2. Is the manuscript technically sound, and do the data support the conclusions?

Reviewer #1: Yes

Reviewer #2: Partly

3. Has the statistical analysis been performed appropriately and rigorously? 

Reviewer #1: Yes

Reviewer #2: (No Response)

4. Have the authors made all data underlying the findings in their manuscript fully available?

Reviewer #1: Yes

Reviewer #2: (No Response)

5. Is the manuscript presented in an intelligible fashion and written in standard English?

Reviewer #1: Yes

Reviewer #2: No

6. Review Comments to the Author

Reviewer #1: Thank you for sending a revised version of the paper. The text has improved significantly in clarity. Tables are also easier to understand in this new version.

Please consider the following edits:

Page 4 line 91: "history" of diabetes

Page 6 line 134: "echolucent" instead of "transparent"

Page 17 line 243: "ductus venosus" instead of "venous catheter"

Reviewer #2: I read with great interest the Manuscript titled “Maternal prenatal screening programs that predict trisomy 21, trisomy 18, and neural tube defects in offspring” (PONE-D-22-00752R1), which falls within the aim of this Journal.

In my honest opinion, methodology is accurate and conclusions are supported by the data analysis.

Nevertheless, authors should clarify some point and improve the discussion citing relevant and novel key articles about the topic.

Authors should consider the following recommendations:

- I suggest to improve the intelligibility and fluidity of the discursive part: in particular the presentation of the results and the conclusions are sometimes ambiguous or not clear (e.g. where there are sentences such as "the differences were statistically significant" should be specificated between which groups).

- I recommend the authors to specify the meaning of the abbreviations before using them (line 59 : NTD),

to correct the word "medium" with "median" (line 141) and to add "age" after "gestational" (line 186)

- I suggest to the Authors to specificate the age range considered "AMA" in table 4 and to add "fetuses" after "healty" at line 338.

- In the Results section, the Authors have simply reported the p values, from which however it is

not possible to deduce the real clinical relevance of the highlighted statistical significance. In

order to better understand the obtained results, I suggest reporting not only the p values, but

also the corresponding confidence intervals

- In the discussion section I could not find considerations about neural tube defects.

- It is important to report the results obtained by the authors in the context of clinical practice

and to adequately highlight the strengths and the contribution this study adds to the literature already existing on

the topic and to future study perspectives.

- Does this manuscript conform the Enhancing the QUAlity and Transparency Of health Research

(EQUATOR) network guidelines? It would be mandatory to declare about this element

- I could not find any information regarding the approval of the Institutional Review Board. Did

author this approval before the study start?

- I recommend to highlight, at least briefly, the role of absolute risk for the detection of fetal

aneuploidies in the first-trimester screening and the impact of advanced maternal age (authors may

refer to: PMID: 27442264; PMID: 25027820).

7. PLOS authors have the option to publish the peer review history of their article (what does this mean?). If published, this will include your full peer review and any attached files.

Reviewer #1: No

Reviewer #2: No

---

## [Author Response · Author response to Decision Letter 1]

24 Dec 2022

Response letter 

Dear editor:

Our paper “Maternal prenatal screening programs that predict trisomy 21, trisomy 18, and neural tube defects in offspring” (PONE-D-22-00752R1), has been modified according to the reviewer's suggestion, and the modified part is marked with yellow shade. 

Reviewer #1: Thank you for sending a revised version of the paper. The text has improved significantly in clarity. Tables are also easier to understand in this new version.

Please consider the following edits:

Page 4 line 91: "history" of diabetes

Page 6 line 134: "echolucent" instead of "transparent"

Page 17 line 243: "ductus venosus" instead of "venous catheter"

Response: Thanks for your advice, we have made modifications to these parts with shaded in yellow.

Reviewer #2: I read with great interest the Manuscript titled “Maternal prenatal screening programs that predict trisomy 21, trisomy 18, and neural tube defects in offspring” (PONE-D-22-00752R1), which falls within the aim of this Journal.

In my honest opinion, methodology is accurate and conclusions are supported by the data analysis.

Nevertheless, authors should clarify some point and improve the discussion citing relevant and novel key articles about the topic.

Authors should consider the following recommendations:

- I suggest to improve the intelligibility and fluidity of the discursive part: in particular the presentation of the results and the conclusions are sometimes ambiguous or not clear (e.g. where there are sentences such as "the differences were statistically significant" should be specificated between which groups).

Response: Thanks for your advice, we have amended the presentation of the results and the conclusions are sometimes ambiguous or not clear.

- I recommend the authors to specify the meaning of the abbreviations before using them (line 59 : NTD),

to correct the word "medium" with "median" (line 141) and to add "age" after "gestational" (line 186)

Response: Thanks for your advice, we have made modifications to these parts with shaded in yellow.

- I suggest to the Authors to specificate the age range considered "AMA" in table 4 and to add "fetuses" after "healty" at line 338.

Response: Thanks for your advice, AMA age range was 35 years or older, we have revised this, and we have added "fetuses" after "healthy" in text.

- In the Results section, the Authors have simply reported the p values, from which however it is not possible to deduce the real clinical relevance of the highlighted statistical significance. In order to better understand the obtained results, I suggest reporting not only the p values, but also the corresponding confidence intervals

 Response: Thanks for your advice, we put confidence intervals in the tables concerned. 

- In the discussion section I could not find considerations about neural tube defects.

Response: Thanks for your advice, we have added considerations about neural tube defects. NTD is a serious neurological defect of the fetus, causing a great challenge to the families and societies [25]. Currently, the diagnosis of ONTD mainly relies on ultrasound images and serum AFP in the second trimester in the maternal serum [25]. Our study showed that the detection rate of NTD in FTS were higher than in ISTS, which may be because that NT examination improved the detection rate of FTS. Therefore, NTD screening should be combined with ultrasound images and maternal serum screening. In addition, Chen et al. [26] Showed that AFP variants (AFP-L2 and AFP-L3) are favorable biomarkers for screening ONTD, and it have superior sensitivity and specificity. Gerardo et al. [27] showed that abnormal intracranial translucency during the first-trimester may be a useful screening marker for early detection of NTDs.

- It is important to report the results obtained by the authors in the context of clinical practice and to adequately highlight the strengths and the contribution this study adds to the literature already existing on the topic and to future study perspectives.

Response: Thanks for your advice, we have added some other viewpoint with shaded in yellow. 

Undeniably, NIPT, as a non-invasive, highly sensitive and specific feature, has become a novel screening method for fetal chromosome malformations. With the rapid development of molecular biology, perhaps one day in the future, the traditional maternal serum biochemical screening method may be a secondary screening method.

Although compared to ISTS, FSTCS did not significantly improve the rate of chromosome disease detection. However, for eligible pregnant women, we still recommend FSTCS, because it will reduce the high risk of chromosomal diseases and reduce the probability of invasive diagnosis in pregnant women.

- Does this manuscript conform the Enhancing the QUAlity and Transparency Of health Research (EQUATOR) network guidelines? It would be mandatory to declare about this element

Response: This article was conformed the Enhancing the QUAlity and Transparency Of health Research (EQUATOR) network guidelines. We have added it in the text.

- I could not find any information regarding the approval of the Institutional Review Board. Did author this approval before the study start?

Response: Thanks for your advice, It is located in the Subjects and Methods.

- I recommend to highlight, at least briefly, the role of absolute risk for the detection of fetal aneuploidies in the first-trimester screening and the impact of advanced maternal age (authors may refer to: PMID: 27442264; PMID: 25027820).

Response: Thanks for your advice, we have added some other statement with shaded in yellow. 

Studies have shown that advanced maternal age is associated with adverse pregnancy outcomes, such as preterm birth, miscarriage, and fetal chromosome abnormalities [28]. In this study, we found that the incidence of trisomy 21 and trisomy 18 was higher in advanced maternal age than in non-advanced maternal age. The detection rate for fetal chromosomal malformations of advanced maternal age in ISTS was higher than that of non-advanced maternal age. But studies have showed that the effect of advanced maternal age on fetal chromosome abnormalities has been overestimated, introducing absolute risk (eliminating the mother's age risk) from the risk algorithm may reduce the screening positive rate for advanced maternal age and increase the screening positive rate for non-advanced maternal age [29].

---

## [Decision Letter · Decision Letter 2]

18 Jan 2023

Maternal prenatal screening programs that predict trisomy 21, trisomy 18, and neural tube defects in offspring

PONE-D-22-00752R2

Dear Dr. Chen,

We’re pleased to inform you that your manuscript has been judged scientifically suitable for publication and will be formally accepted for publication once it meets all outstanding technical requirements.

Kind regards,

Antonio Simone Laganà, M.D., Ph.D.

Academic Editor

PLOS ONE

Additional Editor Comments (optional):

Authors performed the required corrections, which were positively evaluated by the reviewers. I am pleased to accept this paper for publication.

Reviewers' comments:

Reviewer's Responses to Questions

**Comments to the Author**

1. If the authors have adequately addressed your comments raised in a previous round of review and you feel that this manuscript is now acceptable for publication, you may indicate that here to bypass the “Comments to the Author” section, enter your conflict of interest statement in the “Confidential to Editor” section, and submit your "Accept" recommendation.

Reviewer #2: All comments have been addressed

2. Is the manuscript technically sound, and do the data support the conclusions?

Reviewer #2: Yes

3. Has the statistical analysis been performed appropriately and rigorously? 

Reviewer #2: (No Response)

4. Have the authors made all data underlying the findings in their manuscript fully available?

Reviewer #2: Yes

5. Is the manuscript presented in an intelligible fashion and written in standard English?

Reviewer #2: Yes

6. Review Comments to the Author

Reviewer #2: (No Response)

7. PLOS authors have the option to publish the peer review history of their article (what does this mean?). If published, this will include your full peer review and any attached files.

Reviewer #2: No

---

## [Editor Report · Acceptance letter]

9 Feb 2023

PONE-D-22-00752R2 

Maternal prenatal screening programs that predict trisomy 21, trisomy 18, and neural tube defects in offspring 

Dear Dr. Chen:

I'm pleased to inform you that your manuscript has been deemed suitable for publication in PLOS ONE. Congratulations! Your manuscript is now with our production department. 

Kind regards, 

on behalf of

Dr. Antonio Simone Laganà 

Academic Editor

PLOS ONE